# Reduction of Hippocampal High-Frequency Activity in Wag/Rij Rats with a Genetic Predisposition to Absence Epilepsy

**DOI:** 10.3390/diagnostics12112798

**Published:** 2022-11-15

**Authors:** Evgenia Sitnikova, Dmitrii Perevozniuk, Elizaveta Rutskova, Shukhrat Uzakov, Viktor A. Korshunov

**Affiliations:** 1Institute of the Higher Nervous Activity and Neurophysiology of Russian Academy of Sciences, Butlerova Str., 5A, 117485 Moscow, Russia; 2Skolkovo Institute of Science and Technology, Bolshoy Boulevard 30, Bld. 1, 121205 Moscow, Russia

**Keywords:** animal model, spike-wave seizures, preictal period, ictal period, dentate gyrus, RIPPLELAB, cluster analysis

## Abstract

In temporal lobe epilepsy, high frequency oscillations serve as electroencephalographic (EEG) markers of epileptic hippocampal tissue. In contrast, absence epilepsy and other idiopathic epilepsies are known to result from thalamo-cortical abnormalities, with the hippocampus involvement considered to be only indirect. We aimed to uncover the role of the hippocampus in absence epilepsy using a genetic rat model of absence epilepsy (WAG/Rij rats), in which spike-wave discharges (SWDs) appear spontaneously in cortical EEG. We performed simultaneous recordings of local field potential from the hippocampal dentate gyrus using pairs of depth electrodes and epidural cortical EEG in freely moving rats. Hippocampal ripples (100–200 Hz) and high frequency oscillations (HFO, 50–70 Hz) were detected using GUI RIPPLELAB in MatLab (Navarrete et al., 2016). Based on the dynamics of hippocampal ripples, SWDs were divided into three clusters, which might represent different seizure types in reference to the involvement of hippocampal processes. This might underlie impairment of hippocampus-related cognitive processes in some patients with absence epilepsy. A significant reduction to nearly zero-ripple-density was found 4–8 s prior to SWD onset and during 4 s immediately after SWD onset. It follows that hippocampal ripples were not just passively blocked by the onset of SWDs, but they were affected by spike-wave seizure initiation mechanisms. Hippocampal HFO were reduced during the preictal, ictal and postictal periods in comparison to the baseline. Therefore, hippocampal HFO seemed to be blocked with spike-wave seizures. All together, this might underlie impairment of hippocampus-related cognitive processes in some patients with absence epilepsy. Further investigation of processes underlying SWD-related reduction of hippocampal ripples and HFO oscillations may help to predict epileptic attacks and explain cognitive comorbidities in patients with absence epilepsy.

## 1. Introduction

The hippocampus is a part of the limbic system, which plays a key role in memory [1,2,3,4]. Abnormal microstructure and functional disturbances in the hippocampus are known to cause the most common type of epilepsy, i.e., temporal lobe epilepsy [5,6]. On the level of electroencephalogram (EEG) analysis, epileptiform discharges, pathological ripples and high frequency oscillations may serve as EEG markers of epileptic hippocampal tissue [7,8,9,10]. Another type of epilepsy, idiopathic generalized epilepsy (or non-convulsive epilepsy, such as childhood absence epilepsy, juvenile absence epilepsy, juvenile myoclonic epilepsy and epilepsy with generalized grand mal on awakening) are known to be accompanied by generalized spike-wave discharges (SWDs) in EEG [11,12,13]. EEG abnormalities (i.e., SWDs) in idiopathic generalized epilepsies are known to result from abnormalities of the thalamo-cortical neuronal system [14,15,16,17]. Early studies in genetic rat models demonstrated that SWDs have never been recorded in the hippocampus [18,19,20]: “*SWD of the generalized non-convulsive seizures do not spread to the limbic structures in spite of their high excitability*” (page 90 in [18]). Moreover, “*synchronous SWD in absence seizures are restricted to the cortex and lateral nuclei of the thalamus and directly connected structures with the exclusion of the limbic system”* (page 91 in [18]). Finally, Adam Kandel et al. in 1996 published a paper [21], in which they confirmed a lack of hippocampal involvement in a rat model of spike-wave epilepsy.

Later, the role of limbic structures and the hippocampus in generalized non-convulsive epilepsy was reconsidered. In 1998, Athanassios G. Siapas and Matthew A. Wilson demonstrated ~200 Hz functional links between hippocampal ripples and neocortical spindle (7–14 Hz) oscillations [22]. Moreover, an increased synchronization between the hippocampus and the thalamocortical network was found in a pharmacological model of absence seizures, suggesting that the hippocampus was entrained by thalamocortical activity [23,24]. The latter gamma-butyrolactone (GBL) model was considered as a model of atypical SWDs [23,24] in which the hippocampus might be involved, in contrast to typical SWDs, which are ‘purely’ thalamocortical. Filiz Onat et al. published a comprehensive review on mechanisms of typical and atypical absence epilepsy [25], demonstrating that both types of absence epilepsies involved different parts of cortico-thalamo-cortical brain network, and the limbic circuitry seemed to be an integral component of both typical and atypical absence epilepsies.

Here we challenge the idea that the hippocampusis not involved in typical absence epilepsy. We used a genetic rat model of this disorder (WAG/Rij rats) that exhibited spontaneous characteristic SWDs in their cortical EEG and were represented as a valid model [26,27,28]. We analyzed electrical activity in the hippocampal dentate gyrus because this structure is known to be critically involved in limbic epilepsy [8,29]. We made simultaneous recordings of local field potential from the hippocampal dentate gyrus using pairs of depth electrodes and epidural cortical EEG records in freely moving rats. Hippocampal ripples (100–200 Hz) and high frequency oscillations (HFO, 50–70 Hz) were detected using GUI RIPPLELAB in MatLab [30,31]. Here we aimed to check the hypothesis that high-frequency activity in the hippocampus during preictal and ictal periods might be affected by SWD-related mechanisms.

## 2. Materials and Methods

### 2.1. Animals

Experiments in rats were conducted at the Institute of Higher Nervous Activity and Neurophysiology RAS (Moscow, Russia) according to EU Directive 2010/63/EU for animal experiments and approved by our institution’s animal ethics committee (protocol No. 4 from 26 October 2021). Rats were kept in standard conditions with a natural light-dark cycle, and had free access to rat chow and tap water.

### 2.2. Electrodes Implantation and Recording Procedure

Four adult WAG/Rij male rats (8–12 months old; body weight 320–460 g) were used to record electrical brain activity with the set of implanted electrodes. Epidural and intracranial electrodes were put together in a solid plug, containing two depth electrodes, one active epidural electrode, one reference electrode, and a common ground electrode. More specifically, each depth electrode was made of pairwise twisted nichrome wires (wire: enamel insulation, diameter 80 μm, cutting ends of 45 degrees) providing recordings of the local field potential (LFP). Epidural electrodes were made of stainless-steel screws (screw: shaft length = 2.0 mm, head diameter = 2.0 mm, shaft diameter = 0.8mm) providing recordings of the electrocorticogram (ECoG) as shown in Figure 1a. Stereotactic surgery was performed under chloral hydrate anesthesia (325 mg/kg, 4% solution in 0.9% NaCl) in a Kopf stereotaxic apparatus. The surgical area was shaved and treated with a 5% iodine alcohol solution. In addition, a 2% lidocaine solution was used for local infiltration anesthesia. Hair, soft tissues, and periosteum were removed.

One depth bipolar electrode was inserted into the dentate gyrus (AP −3.2; L 1.8; H 3.35–3.40. All coordinates are given in mm relative to the bregma) using electrophysiological control for precise positioning. A pair of nichrome electrodes was temporally inserted into the right perforate pathway (AP −6.8; L 3.4; H 2.25–2.95), and electrical stimulation of the perforant pathway during the surgery was used to find the best position of the hippocampal electrode (this position was fixed when the response amplitude was maximal). Electrodes from the perforant pathway were removed, and the holes were closed with a methacrylate monomer. An active epidural screw electrode was implanted over the right frontal cortex (AP +2; L 2.5) and the reference screw electrode was placed over the right cerebellum. A common ground screw electrode was placed in the left hemisphere (AP −9.0–9.3; L 4.0–4.5). Another depth bipolar electrode was inserted into the ventrobasal region of the right thalamus (AP −3.2; L 2.4–2.6; H 5.8–6.0). Thalamic signal carried supplementary information needed for identification of states (synchronized/desynchronized), evaluation of locality of high-frequency oscillations and detection of movement artifacts. The electrode’s plug was permanently fixed to the skull with a methacrylate monomer. The thalamic signal per se was not analyzed in the present study. After the surgery, the rats were allowed to recover for 5–6 days.

EEG recording was performed in freely moving rats placed in Plexiglas cages located in an electromagnetically shielded camera. The electrode plug on the rat’s head was connected to the amplifier via flexible wire and a swivel contact. Signals were fed into a differential biopotential amplifier NBL304 (Neurobiolab, Moscow, Russia). Intracranial signals (from paired bipolar depth electrodes) were recorded throughout the two-wire connection preamplifier [32]. The epidural signal was recorded with a direct differential connection to the amplifier. Data was acquired with L-Card ADC (L-Card, Russia) and PowerGraph software (PowerGraph, Moscow, Russia). The sampling frequency was 1000 Hz per channel, and a 50 Hz notch filter and a high-pass filter at 1 Hz were used.

We performed recordings twice for each animal (except for the rat G10 due to technical issues). Recording sessions were separated by several hours/days. After the end of recording sessions, the rats were sacrificed by a lethal dose of chloral hydrate solution. Electrical current was applied through depth electrodes (5 mA for 10 s) to make electrical lesions. The brains were removed and fixed in buffered 4% formaldehyde (pH = 7.0) for 5–7 days. Then brains were kept in a 30% sucrose solution in a phosphate buffer (pH = 7.0) for cryoprotection (2–3 days, t = 4 °C). Serial coronal slices 20 μm thick were made with a microtome cryostat (CTB 6, Medite GmbH, Burgdorf Germany), mounted on glass microscope slides (Superfrost^®^Plus Gold, Menzel GmbH, Braunschweig, Germany), and stained with 0.1% cresyl violet. In brain slices, the location of electrodes was determined according to Paxinos’ rat brain [33].

### 2.3. Automatic Recognition of Hippocampal Oscillations

Hippocampal ripples (see an example of ripples in Figure 1a) were detected automatically using the Short Line Length detector algorithm developed by Andrew Gardner et al. [34]. Briefly, the raw LFP signal was band-pass filtered at 100–200 Hz, and the energy of the signal, E(t), was calculated by a short time line length (SLL) measure using the formula
(1)$$E\left(t\right)=\overset{t}{\underset{k}{\sum }}|x_{k}-x_{k-1}|$$
where t is the time moment (ms); k is the time moment defined as k = t − N + 2 with window N = 5 ms, and x_k_ is the value of the amplitude at the time moment k. The signal was divided into 3-min epochs, and the empirical cumulative distribution function was obtained for each epoch. An event was identified as a ripple if its SLL measurement was greater than the 97.5th percentile of the empirical cumulative distribution function for each 3-min epoch and if it had a minimum duration of 12 ms. The results of automatic detections were checked manually for the presence of error detections. Figure 1b shows an example of a ripple detected in the hippocampal LPF using RIPPLELAB.

Hippocampal high frequency oscillations (50–70 Hz, HFO) were detected automatically using the Short Time Energy (STE) algorithm proposed by Staba et al. [35]. This method demonstrated better performance for selecting HFO in comparison to the abovementioned SLL method. The raw LFP signal was band-filtered at 50–70 Hz, and the STE of the signal, E(t), was calculated using the root mean square measure, RMS, by the equation:(2)$$E\left(t\right)=\sqrt[{2}]{\frac{1}{N}\overset{i}{\underset{k=t-N+1}{\sum }}x_{k}^{2}}$$
where t is the time moment (ms), and N = 3 ms time window. An event was identified as an HFO if its successive RMS values were greater than 1 standard deviation (SD) above the overall RMS mean, contained more than 20 peaks with amplitudes above 1 SD and lasted longer than 5 ms. The minimal interval between successive HFOs was set to 300 ms to prevent overlap. Figure 1c shows an example of an HFO detected in the hippocampal LPF using RIPPLELAB. The results of automatic detections were checked manually for the presence of error detections. Considering that HFOs lasted longer than ripples and minimum time interval between the nearest HFO was 300 ms, the results of HFO detections were represented as discrete data that took the value of 0 (no HFO) or 1 (HFO presence) at 500 ms time intervals.

### 2.4. Temporal Dynamics of Hippocampal Oscillations

To score hippocampal oscillations in the baseline, ripples and HFO were detected during the interictal period in artifact-free epochs with a minimum 5 min duration more than 10 min before and 10 min after SWDs. Computation in baseline periods was performed during the state of brain synchronization and desynchronization (Section 2.3 and Figure 2a,b). Then, we scored hippocampal oscillations during the preictal, ictal and postictal periods (Figure 2c) using slightly different approaches.

Ripples were detected during the preictal and ictal periods with a 2 s bin size. During the preictal period, ripples were scored in the following consecutive intervals: (1) from 120 s to 60 s before SWD onset; (2) from 60 s to 30 s before SWD onset; (3) from 30 s to 0 s before SWD onset, and (4) during additional short interval from 15 s to 0 s before SWD onset (Figure 3a). During the ictal period, ripples were detected from 0 s to 8 s after the onset of SWDs (Figure 3b). Data are presented as quantitative discrete attributes. For all the intervals, ripple density was defined as a ratio of the number of ripples in the given interval to the interval’s length.

Analysis of HFOs was performed using a 0.5 s bin, including 5 s periods before SWD onset, 5 s after SWD onset and 5 s before SWDs end (ictal period), and 5 s after SWDs ended (postictal period, Figure 4). The baseline level was assessed during the desynchronized state of equal duration, i.e., 600-s periods (30 epochs of 20 s = 1200 windows of 0.5 s duration).

Statistical analysis included one-factor, two-factors and repeated measures ANOVA for normally distributed variables, Friedman ANOVA for non-normally distributed variables, a Bonferroni test for post-hoc analysis, random trees classification, and K-means clustering. Details of statistical tests can be found in the Results section.

## 3. Results

Recording sessions lasted from 0.99 to 3.37 h. One rat (G12) had no SWDs during two sessions of 0.99 h and 1 h duration. Three other rats showed from 7.7 to 60.2 SWDs per hour (see an example of SWDs in Figure 1c). Ripples and HFO were found during desynchronized states (Figure 2a) and synchronized states (Figure 2b). Mean ripple density during the synchronized state was around 0.088 per second (minimum 0.005 and maximum 0.286) and, during the desynchronized state, around 0.070 per second (minimum 0.009 and maximum 0.240). Mean HFO density during the synchronized state was 0.44 per second (minimum 0.30 and maximum 0.48) and, during the desynchronized state, around 0.33 per second on average (minimum 0.212 and maximum 0.425). Density of HFO was higher than density of ripples (ANOVA, the factor ‘oscillation type’ HFO/ripple was significant F_1;22_ = 52.1 *p* < 0.001), but no differences were found between states (ANOVA, the factor ‘state’ synchronized/desynchronized was not significant F_1;22_ = 2.3 *p* = 0.14). Data are shown in Table A1.

### 3.1. Ripples

Dynamics of ripple activity were assessed using three approaches. The first and second approaches were based on cluster analysis of ripple distributions. We hypothesized that hippocampal ripples before and during SWDs represent statistically different groups. We began with random trees classification to estimate the number of clusters. This indicated the presence of three distinctive clusters (the exploratory analysis was not shown). Next, these clusters were verified with K-means clustering.

First, the preictal period 2 min immediately prior to the onset of SWDs was analyzed (cluster analysis) in 1 min, 30 s and 15 s time windows (Figure 3a). K-means cluster analysis of ripple density was performed in all 297 SWDs, including 207 SWDs with ripples and 90 SWDs without ripples, during the analyzed 120-s interval. Three statistically different clusters were defined (ANOVA for all four intervals F_2;294_ = 417 ÷ 853, all *p*’s < 0.00001, Figure 3a). Two-factors repeated measures ANOVA for the factor ‘cluster’ was F_2;294_ = 1787, *p* < 0.0001; for the factor ‘period’ − F_3;882_ = 33.1, *p* < 0.0001; ‘cluster’*‘period’ interaction F_6;882_ = 27.7, *p* < 0.0001. The majority of cases fell into Cluster 3 (258 SWDs or 86.9% of all SWDs) and showed near-zero ripple density and differed from Cluster 1 and Cluster 2 (*p* < 0.0001, Bonferroni post-hoc tests, Figure 3a). Cluster 1 (14 SWDs or 4.7% from all SWDs) and Cluster 2 (25 SWDs or 8.4% from all SWDs) were characterized by significantly higher ripple density during the 120s interval than Cluster 3. Therefore, only 13.6% of SWDs were preceded by hippocampal ripples during a 2 min preictal period, and there were two significantly different clusters characterized by different preictal dynamics. Each cluster contained measures from different rats; therefore, clusters did not represent individual differences in ripple activity. Rather spike-wave seizures could represent at least three seizure types in reference to the involvement of hippocampal processes.Second, a 40 s period was analyzed (cluster analysis) with a 2 s bin size from −30 s before the onset of SWDs (0 s) to 10 s after the onset of SWDs (Figure 4). Using K-means cluster analysis in all 297 SWDs, we defined three statistically different clusters (ANOVA for all twelve intervals F_2;294_ = 10.7 ÷ 135.5, all *p*’s < 0.0001). Again, the majority of cases fell into Cluster 3 with near zero-ripple-density (268 SWDs or 88.5% of all SWDs). Cluster 1 (15 SWDs, 5.1%) and Cluster 2 (19 SWDs, 6.4%) showed higher ripple density and slightly different dynamics (Bonferroni post-hoc test *p* < 0.05, Figure 3b).Third, ripple density was analyzed in preictal, ictal/postictal periods with a 2 s bin size during the period −16 to 8 s around the SWD start time (i.e., a 24 s period, Figure 4). During this period, hippocampal ripples were found in 58.9% of SWD-containing periods (*n* = 175), and the remaining 41.1% of periods showed no ripples (*n* = 122). In the first group, hippocampal ripples during the 24-s period were distributed irregularly (Friedman ANOVA, χ^2^_Friedman_ (df = 12) = 22.1, *p* = 0.036). The Bonferroni test indicated that ripple density did not differ from zero during two 2 s preictal periods −8 to −4 s, and immediately after SWDs onset 0 to 4 s (*p* < 0.05, Figure 5). Therefore, a significant reduction to nearly zero-ripple-density was found 4–8 s prior to SWDs onset and during the 4 s immediately after SWDs onset.

### 3.2. 50–70 Hz High-Frequency Oscillations

Analysis of HFO was done in 30 SWD-containing periods in each subject (29 SWDs in rat G10). HFO density during baseline was significantly higher than during SWDs-containing periods (ANOVA, factor ‘Period’ F_4;591_ = 22.6, *p* < 0.0001, Bonferroni post-hoc test *p* < 0.0001, Figure 6a). Analysis of HFO density per 0.5 s bin (Figure 6b) indicated that HFOs were diminished during ictal and postictal periods. In other words, hippocampal HFOs were blocked during and after spike-wave seizures.

## 4. Discussion

We challenged the idea that the hippocampus is not involved in typical absence epilepsy by describing a correlation between hippocampal high frequency activity and SWD appearance in WAG/Rij rats. In earlier studies of hippocampal activity in WAG/Rij rats, hippocampal signals were recorded with a pair of electrodes placed in the hippocampus and the cerebellum [36,37]. Post-mortem histological control was used to confirm hippocampal electrode actual location. Here we recorded hippocampal local field potential using a pair of twisted nichrome wires with 80 μm diameter implanted in the dentate gyrus (intracranial LFP). During the surgery we controlled electrode positioning in the dentate gyrus by monitoring hippocampal response to the perforant pathway stimulation. By this means we achieved a greater locality and therefore precision of the recorded hippocampal signals.

We detected ripples and HFO using an analytical approach by Miguel Navarrete et al. implemented in GUI RIPPLELAB for MatLab [30,31]. These specific patterns required different algorithms of detection: a Short-time Line Length (SLL) algorithm [34] for ripple detection and a Short Time Energy (STE) algorithm [35] for HFO detection. Neither method required complex computations, and could be applied quickly and easily. At the same time, both methods relied upon band-path filtering that is appropriate for detecting high-amplitude events such as ripples and HFO. In particular, STE [35] is more efficient for detecting very short oscillations, such as ripples, while SLL [34] allows precise identification of the beginning and the end of longer oscillations, such as HFOs.

Both ripples and 50–70 Hz oscillations were found in healthy periods during synchronized and desynchronized states. The density of 50–70 Hz oscillations was higher than the density of ripples in both states, with no differences between states. We concluded that the hippocampus more readily generated 50–70 Hz oscillations than ripples.

The present study can be characterized as a pilot study done in four rats. The group size could be larger, but the surgery was rather invasive, and we reduced the number of subjects to a minimum. These were subjects with strong-to-moderate epilepsy and one without epilepsy. Our results indicated that hippocampal ripples and HFOs occurred rarely. Hippocampal ripples were found in only 13.6% of cases when inspecting 2 min preictal SWDs periods. When SWDs were accompanied by hippocampal ripples, a significant reduction to nearly zero-ripple-density was found 4–8 s prior to SWDs onset and during 4 s immediately after SWDs onset. Thus, hippocampal ripples seemed to be not just passively blocked by the onset of SWDs, but were affected by spike-wave seizure initiation mechanisms. Interestingly, based on the dynamics of hippocampal ripples, SWDs were divided into three clusters, possibly representing at least three seizure types in reference to the involvement of hippocampal processes. Further analysis of certain SWDs could clarify this issue. Hippocampal HFOs were reduced during the preictal, ictal and postictal periods in comparison to the baseline, and seemed to be blocked by spike-wave seizures generation. Investigation of processes underlying SWD-related reduction of hippocampal ripples and HFO oscillations may help to find new ways to control epileptic attacks.

Considering associations between hippocampal ripples and cognitive processes [2,3], we assumed that SWD-related dynamics of ripples might be connected to observed cognitive deficits in WAG/Rij rats [36,37]. Similarly, this possibly could explain cognitive comorbidities in patients with absence epilepsy, and this might be used as an additional biomarker characterizing epileptic profile. For example, a sharper decrease in ripples might be found in patients with severe forms of absence epilepsy.

## 5. Conclusions

In the current study we demonstrated that certain alterations of hippocampal activity accompanied epileptic spike-wave discharges in WAG/Rij rats. Hence, we obtained evidence that hippocampal functioning is affected by absence epilepsy. High-frequency activity in the hippocampus was either blocked or decreased before, during and immediately after spike-wave discharges. The processes underlying SWD-related reduction of hippocampal ripples and high-frequency oscillations requires further investigation and may lead to development of a new approach in preventing epileptic attacks. The obtained results allow us to hypothesize a link between SWD-related reduction of high-frequency hippocampal activity and cognitive comorbidities in some patients with absence epilepsy.

## Figures and Tables

**Figure 1 diagnostics-12-02798-f001:**
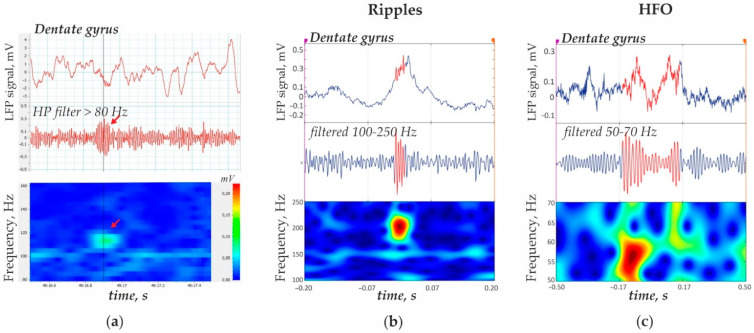
Representative hippocampal ripples and 50–70 Hz high frequency oscillations (HFO) in time and frequency domains. (**a**) Ripple oscillation in the dentate gyrus LPF could be visualized in the filtered signal and in the time-frequency plot (red arrow). The amplitude spectrum was plotted with a fast Fourier transform (length 512, window overlap 93.75%). (**b**) Ripple oscillation detected by means in RIPPLELAB. (**c**) High frequency oscillation 50–70 Hz (HFO) detected by means of RIPPLELAB.

**Figure 2 diagnostics-12-02798-f002:**
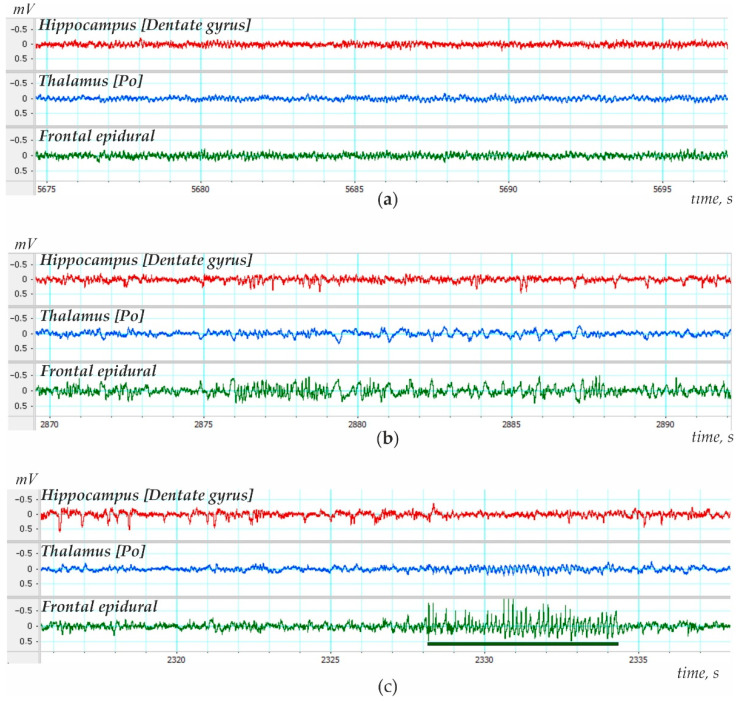
Electrical brain activity recorded in adult WAG/Rij rats with free behavior. (**a**) LPF/ECoG recording made during the desynchronized brain state characterized by low amplitude fast activity. (**b**) LPF/ECoG recording made during the synchronized brain state characterized by high amplitude slow wave activity. (**c**) LPF/ECoG recording during spontaneous spike-wave discharges (SWDs, underlined).

**Figure 3 diagnostics-12-02798-f003:**
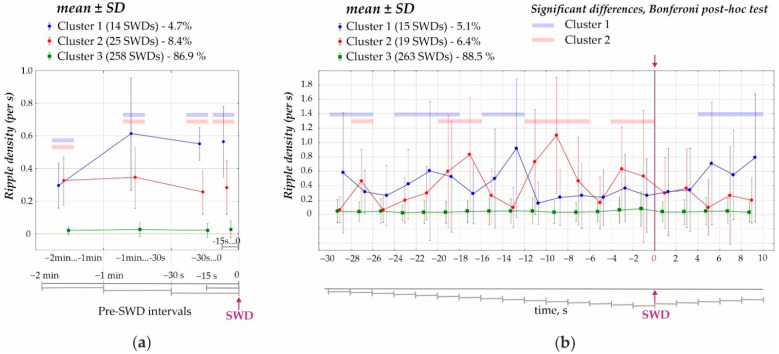
Density of hippocampal ripples before and during SWD. (**a**) Ripple density during the 120 s preictal period divided into three statistically different clusters. Blue and rose stripes highlight significant differences of Cluster 1 (4.7% from all SWDs, shown in blue) and Cluster 2 (8.4%, shown in rose) from Cluster 3 (86.9%, shown in green, all *p*’s < 0.0001, Bonferroni tests. (**b**) Ripple density during a 40 s period around the onset of SWDs divided into three statistically different clusters. Blue and rose stripes mark significant differences between Cluster 3 and Cluster 1 (blue) & Cluster 3 and Cluster 2 (rose) (all *p*’s <0.05 Bonferroni tests).

**Figure 4 diagnostics-12-02798-f004:**
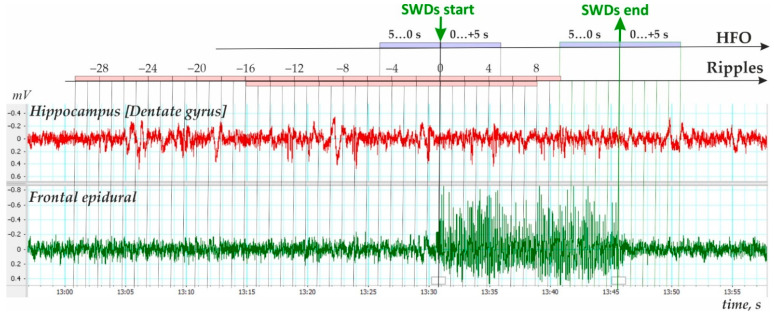
Schema representing the epochs when hippocampal ripples and high-frequency oscillations (HFO) were analyzed. Arrows indicate the onset and end of spike-wave discharges (SWDs).

**Figure 5 diagnostics-12-02798-f005:**
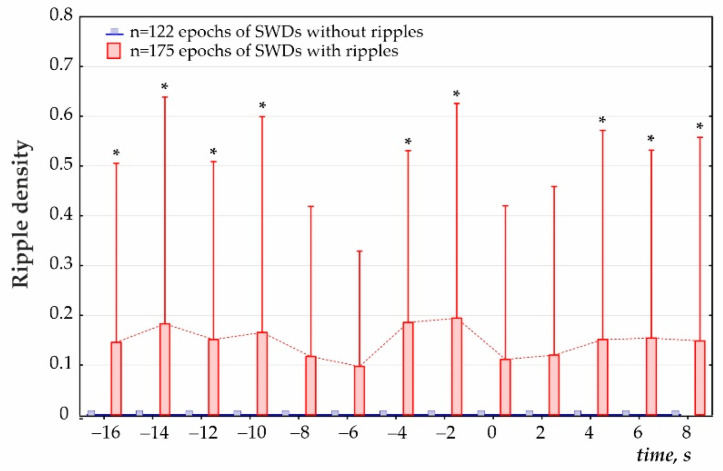
Dynamics of ripple density before and during SWDs. The onset of SWDs is zero time. *—significant differences between the SWD-containing epochs with ripples (*n* = 175) and SWDs without ripples (*n* = 122), Bonferroni test, *p* < 0.05.

**Figure 6 diagnostics-12-02798-f006:**
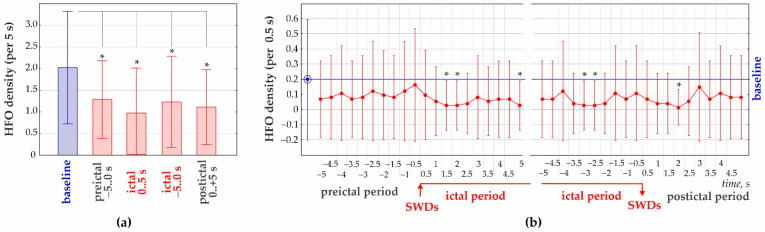
Density of hippocampal 50–70 Hz high-frequency oscillations (HFO) in relation to the presence of spike-wave discharges (SWDs). (**a**) Density of HFOs as measured in 5 s periods; * significant differences from the baseline (the desynchronized state), Bonferroni test, *p* < 0.05. (**b**) Dynamics of HFO density before, during and after SWDs assessed per 0.5 s; * significant differences from baseline, Bonferroni test, *p* < 0.05.

## Data Availability

The data presented in this study are available on request from the corresponding author for further research.

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
