# Peer review of "Reduction of Hippocampal High-Frequency Activity in Wag/Rij Rats with a Genetic Predisposition to Absence Epilepsy"

_diagnostics, 2022, doi:10.3390/diagnostics12112798_

Round 1
Reviewer 1 Report
The manuscript by Sitnikova et al. describes changes in hippocampal oscillatory activity that occurs in the WAG/Rij rats, a model of absence epilepsy, which displays spontaneous cortical spike-wave discharges (SWDs). Adult WGA/Rji rats were implanted with an epidural and two intracranial electrodes (one on the thalamus and other in the hippocampus) and electrophysiological analyses of freely moving rats performed during two recording session of about 1 hr each. They used the GUI RIPPLELAB to detect high frequency oscillations recorded from intracranial electrodes placed in the dentate gyrus and used two algorithms, the short life length (SLL) and the short time energy (STE), for recognition of ripples (100-200 Hz) and high frequency oscillations (HFO, 50-70 Hz), respectively. They found that SWDs blocked hippocampal HFO as these oscillations were reduced during preictal, ictal and postictal, while ripples were not blocked but reduced only at 5-4 seconds prior and after SWD onset.
This is an interesting study and with a good experimental design to evaluate how the SWDs affect hippocampal oscillations. However, the text could be improved for clarity.
1) The rationale for using cluster analysis of ripples activity is unclear, specially because the authors do not use the information in the discussion. Also, how clusters were defined is missing and there is lack of information about the relevance of hippocampal ripples (and HFO) to cognition.
2) No information is given regarding the recordings performed on the thalamus. Where they analyzed? If not, why were they implanted?
3) Is there any difference among the SWDs which could explain why only some affect hippocampal ripples? The authors state that they are different seizure types, why?
4) The aim of the study could be better stated. The same applies to the conclusion/discussion that does not seem to indicate whether the goal was attained. On line 280 it is stated that “we have highlighted the uncertainty”. Is that so, or the authors mean “clarified the controversy?”
Other comments.
1) Fig. 1a, the illustrations are very small and do not provide any useful information. Figs 1b-d, the axis values are unreadable and the markers for SDW are invisible.
2) Lines 134-135, should be pH and not ph.
3) Line 142, Example of ripples instead of exemplary ripples.
4) In Methods, the description of the time interval between the two session recording is missing.
5) Line 155, Representative and not representation.
6) Provide a general description of statistical analyses performed on line 197.
7) Line 213, “ripple activity was assessed” instead of ripple activity has been accessed.
8) Define cluster analysis method used.
9) Fig. 3b, it is unclear which color of the statistical stripes correspond to differences between the clusters (which pairwise comparison?).
10) Table A1. It would be better adding horizontal lines to separate the rats. Why rat G10 was recorded for one session only while the others were recorded in two sessions?
11) Table A2. Average for the number of SWDs is incorrect. The value there is the sum.
Author Response
Dear Reviewer,
Thank you for your positive decision on our manuscript and for highlighting the strong points of our study. We highly appreciate the time and effort that you dedicated to evaluating our paper. We revised it in accordance with your critical remarks. First, the aim and the rationale were revised. Second, some technical and methodological details were explained in more detail. Third, the illustrations were corrected. English has also been corrected, because moderate English changes were required. Please find our point-by-point answers below.
1) The rationale for using cluster analysis of ripples activity is unclear, specially because the authors do not use the information in the discussion. Also, how clusters were defined is missing and there is lack of information about the relevance of hippocampal ripples (and HFO) to cognition.
Response. Thank you for emphasizing these critical issues. We revised our paper and clarified the aim (Introduction) and the rationale for using cluster analysis (Page 6). “We hypothesized that hippocampal ripples before and during SWDs represent statistically different groups. We began with random trees classification in order to estimate the number of clusters. It gave a clue to the presence of three distinctive clusters (this exploratory analysis was not shown). Next, these clusters were verified with K-means clustering”.
The relevance of hippocampal ripples (and HFO) to cognition is discussed on Page 10: “Considering associations between hippocampal ripples and cognitive processes [2,3]...”.
2) No information is given regarding the recordings performed on the thalamus. Where they analyzed? If not, why were they implanted?
Response. In this study, the thalamic signal carried supplementary information needed for identification of states (synchronized/desynhronized), for evaluation of locality of high-frequency oscillations and detection of movement artifacts. Thalamic signal per se has not been analyzed in the present study. We added this information on Page 3.
3) Is there any difference among the SWDs which could explain why only some affect hippocampal ripples? The authors state that they are different seizure types, why?
Response. It is a fascinating idea. When we defined three clusters, we retrospectively evaluated SWDs for the presence of specific properties. We did not find any relevant differences between SWDs that could potentially explain why some SWDs were associated with certain dynamics of hippocampal ripples.
4) The aim of the study could be better stated. The same applies to the conclusion/discussion that does not seem to indicate whether the goal was attained. On line 280 it is stated that “we have highlighted the uncertainty”. Is that so, or the authors mean “clarified the controversy?”
Response. We revised the Introduction and better formulated the objectives. Page 2: “Here we have challenged the idea that the hippocampus was not involved in typical absence epilepsy … Here we aimed to evaluate the hypothesis that high-frequency activity in the hippocampus during preictal and ictal periods might be affected by SWD-related mechanisms.”
Statements about “the uncertainty” and “clarified the controversy” were deleted.
Other comments.
1) Fig. 1a, the illustrations are very small and do not provide any useful information. Figs 1b-d, the axis values are unreadable and the markers for SDW are invisible.
Response. We updated this figure. Now it is Fig.2. Part (a) was deleted. In the rest parts, axis values were enlarged, and SWD is now clearly marked.
2) Lines 134-135, should be pH and not ph.
Response. Corrected.
3) Line 142, Example of ripples instead of exemplary ripples.
Response. Corrected.
4) In Methods, the description of the time interval between the two session recording is missing.
Response. On page 3 (Method), we describe the time interval between the two recording sessions. “We performed recording twice for each animal (except for the rat G10 due to technical issues); recording sessions were separated by several hours/days.”
5) Line 155, Representative and not representation.
Response. Corrected.
6) Provide a general description of statistical analyses performed on line 197.
Response. We used different statistical methods for evaluating different data. In the revised version, we added (Page 5): “Statistical analysis included one-factor, two-factors and repeated measures ANOVA for normally distributed variables, Friedman ANOVA for non-normally distributed variables, a Bonferroni test for post-hoc analysis, random trees classification, K-means clustering. Details of statistical tests could be found in the Results section.”
7) Line 213, “ripple activity was assessed” instead of ripple activity has been accessed.
Response. Corrected.
8) Define cluster analysis method used.
Response. We added the rationale and more details on cluster analysis in the “Results” section (Page 7). "We began with random trees classification in order to estimate the number of clusters. It gave a clue to the presence of three distinctive clusters (this exploratory analysis was not shown). Next, these clusters were verified with K-means clustering."
9) Fig. 3b, it is unclear which color of the statistical stripes correspond to differences between the clusters (which pairwise comparison?).
Response. Figure legend has been updated to clarify this issue. It says: “Blue and rose stripes mark significant differences between Cluster 3 and Cluster 1 (blue) & Cluster 3 and Cluster 2 (rose) ”
10) Table A1. It would be better adding horizontal lines to separate the rats. Why rat G10 was recorded for one session only while the others were recorded in two sessions?
Response. For table formatting, we used the templates provided by Journal. Now we have added horizontal lines to separate rats in Table A1. On page 3 we mentioned that only one recording session was performed “in the rat G10 due to technical issues”.
11) Table A2. Average for the number of SWDs is incorrect. The value there is the sum.
Response. True. It was the sum of the number of SWDs. This information seemed excessive, and was removed from Table A2.
Reviewer 2 Report
The aim of this study was to understand the interaction between hippocampus and cortex in absence epilepsy using the rat model WAG/Rij. The authors were able to optimise simultaneous recording of intracranial LFP in hippocampal dentate gyrus and cortical epidural EEG in freely moving rats, a challenging technique that warrants recognition. The methods were well-explained. Furthermore, the authors applied RIPPLELAB to detect ripples and HFO, and found that both were affected by SWDs especially at the onset, which could potentially be used as biomarkers for absence epilepsy. Overall, this is an interesting study that contributes to the field.
However, this seems more like a pilot study as the authors only used four rats, with one not showing SWDs for the duration of the recording, which also explains the large standard error. What is the minimum group size based on observed effect size using power analysis?
Absence epilepsy worsens over time if undiagnosed and untreated. Do the authors speculate an age-related change in hippocampal-cortical interaction?
Do the authors think AEDs might have an impact on hippocampal ripples and HFO? Can hippocampal-cortical interaction be used to predict the efficacy of an AED?
It remains unclear if these hippocampal changes during SWDs are the cause of, or lead to, hippocampal-mediated cognitive deficits, as behavioural assays were not performed. While the authors touched on this, it should be highlighted more in future directions.
Other minor comments:
- Figures not cited according to order of appearance (for example Figs 2-4 comes before Fig 1b-d, Fig 1c comes before Fig 1b, etc).
- Figure 1 (b) is slightly cropped at the top.
- Figure 1 (d) s1 and s0 markers are difficult to see, perhaps a red line or arrow to better indicate?
Author Response
Dear Reviewer,
Thank you for your positive decision on our manuscript. We highly appreciate the time and effort that you dedicated to evaluating our paper. Your critical remarks helped us to emphasize practical application of our study. We also mentioned that it was a pilot study and discussed putative hippocampal-mediated cognitive deficits. Your minor comments helped us figure out how to improve the illustrations. Please find our point-to-point answers below.
However, this seems more like a pilot study as the authors only used four rats, with one not showing SWDs for the duration of the recording, which also explains the large standard error. What is the minimum group size based on observed effect size using power analysis?
Response. It is indeed a pilot study in four rats. The group size could be larger but the surgery was rather invasive, and we reduced the number of subjects to a minimum. These were subjects with strong-to-moderate epilepsy and one without epilepsy. Our results indicated that hippocampal ripples and HFO occurred rarely, and the effect of absence epilepsy on ripples/HFO was relatively weak. Therefore, we do not see a reason to repeat experiments in a larger group of rats. We discussed this issue on Page 10.
Absence epilepsy worsens over time if undiagnosed and untreated. Do the authors speculate an age-related change in hippocampal-cortical interaction?
Response. There should be age-related changes in hippocampal-cortical interactions that associate with age-related increase of absence epilepsy in our subjects. The problem of age-related changes in hippocampal and cortical networks is very complex. In fact, SWDs are absent in WAG/Rij rats before 3-4 months of age, and epilptogenesis in WAG/Rij rats seemingly link to aging mechanisms. In all, we are not ready to speculate on age-related changes in hippocampal-cortical interaction in the framework of our study.
Do the authors think AEDs might have an impact on hippocampal ripples and HFO? Can hippocampal-cortical interaction be used to predict the efficacy of an AED?
Response. This question relates to a practical application of our study. The role of hippocampal-cortical interactions in the epileptogenesis of absence epilepsy is unexplored, and we have no data to discuss the effect of anti-absence drugs. Instead, we concluded (Page 11): “Investigation of processes underlying SWDs-related reduction of hippocampal ripples and HFO oscillations may help to find new ways to control epileptic attacks.”
It remains unclear if these hippocampal changes during SWDs are the cause of, or lead to, hippocampal-mediated cognitive deficits, as behavioural assays were not performed. While the authors touched on this, it should be highlighted more in future directions.
Response. This is another practical issue. We added in Discussion the ideas about cognitive deficit in epileptic WAG/Rij rats (Page 10). “... observed cognitive deficits in WAG/Rij rats [36,37]... this possibly could explain cognitive comorbidities in patients with absence epilepsy…”
Other minor comments:
- Figures not cited according to order of appearance (for example Figs 2-4 comes before Fig 1b-d, Fig 1c comes before Fig 1b, etc).
Response. We changed the order of the figures.
- Figure 1 (b) is slightly cropped at the top.
Response. Corrected.
- Figure 1 (d) s1 and s0 markers are difficult to see, perhaps a red line or arrow to better indicate?
Response. We totally updated this figure. Now it is Fig.2. Axis values were enlarged, and the SWD has now been clearly marked.